# Drivers of Antibiotic Use in Semi-Intensive Poultry Farms: Evidence from a Survey in Senegal

**DOI:** 10.3390/antibiotics12030460

**Published:** 2023-02-24

**Authors:** Eve Emes, Adiouma Faye, Nichola Naylor, Dagim Belay, Babacar Ngom, Awa Gueye Fall, Gwen Knight, Michel Dione

**Affiliations:** 1Centre for the Mathematical Modelling of Infectious Diseases, London School of Hygiene and Tropical Medicine, Keppel Street, London WC1E 7HT, UK; 2International Livestock Research Institute, Rue 18 Cité Mamelles, Dakar BP 24265, Senegal; 3UK Health Security Agency, 61 Colindale Av., London NW9 5EQ, UK; 4Department of Infectious Disease Epidemiology, Faculty of Epidemiology and Population Health, Centre for Antibiotic Resistance, London School of Hygiene and Tropical Medicine, Keppel Street, London WC1E 7HT, UK; 5Department of Food and Resource Economics, University of Copenhagen, Nørregade 10, 1165 Copenhagen, Denmark; 6Veterinary Services Directorate, Ministry of Livestock and Animal Products of the Republic of Senegal, 37 Avenue Pasteur, Dakar BP 67, Senegal

**Keywords:** antimicrobial resistance, antimicrobial stewardship, One Health, agriculture, biosecurity

## Abstract

Antimicrobial resistance (AMR), the capacity of microbial pathogens to survive in the presence of antimicrobials, is considered one of the greatest threats to human health worldwide and is growing rapidly in importance. AMR is thought to be driven in part by the use of antimicrobials (AMU) in livestock production. AMU reduction in agriculture is therefore important, but doing so may endanger farmers’ livelihoods and hamper broader food security. Understanding the drivers for farmers’ antibiotics use is essential for designing interventions which avoid harming agricultural output and to safeguard farmers’ economic security. In this study, we analyse AMUSE survey data from poultry farmers in Senegal to explore the effects of vaccination, attitudes towards AMR, and biosecurity practices on: AMU, animal mortality, and farm productivity. We found that farmers with more “AMR-aware” attitudes may be less likely to use antibiotics in healthy birds. Stronger on-farm biosecurity was associated with less use of antibiotics in healthy birds, and in some specifications was linked to higher broiler productivity. Vaccination and AMU were both higher in farms with a higher disease prevalence, and both factors appeared conducive to higher broiler productivity. Overall, there is evidence that awareness raising and biosecurity improvements could encourage prudent use of antibiotics, and that biosecurity and vaccination could to some extent replace antibiotic use as productivity-enhancing and disease management tools in broiler farms. Finally, issues of farm antimicrobial stewardship must be considered at the structural level, with farm behaviours contingent on interaction with state and private stakeholders.

## 1. Introduction

Antimicrobial resistance (AMR), the capacity of microbial pathogens to survive in the presence of antimicrobials, is considered one of the greatest threats to human health worldwide and is growing rapidly in importance [1,2]. Although AMR has always existed, its increasing prevalence is driven largely by the use of antimicrobials (AMU) by humans [3]. In particular, use of antibiotics in livestock animal production is one of the biggest contributors to total AMU, and reducing its use has been identified as a policy priority [4,5,6,7]. As a middle-income country with a high rate of economic growth, Senegal is identified as suffering from the ‘double-burden’ of rising antibiotic availability and meat consumption, combined with rates of bacterial infections that remain high in the global context [1].

Senegal’s most recent National Action Plan on AMR involved the animal health and food safety sectors [8], and aims to balance rational use of antibiotics and awareness raising on AMR with infection control across all One Health sectors. These findings, and others, will contribute to the evidence base which feeds into the upcoming 2023–2027 plan.

Much antibiotic use globally is deemed to be unnecessary or irrational: for example, antibiotics are commonly used as agricultural growth promotors, or are used purely prophylactically (preventatively) rather than therapeutically, and may often be used without a prescription [9]. However, antibiotics can play a therapeutic role in livestock production, and even sub-inhibitory and non-therapeutic use can play a role in animal productivity, and may thus be important to farmers’ income security [9]. Therefore, reducing AMU in livestock production, especially in small-scale and semi-intensive farms, may harm farmers’ livelihoods and economic security, and may contribute to food insecurity at the population level if it negatively affects farm productivity. Achieving a reduction in farm AMU will not be realistic or safe if farmers do not feel secure in doing so. It is therefore important to understand which interventions can be paired with AMU reduction that can prevent any associated loss in farm productivity, and can make farmers feel more comfortable withdrawing or replacing antibiotics.

We investigated this question using the case study of semi-intensive peri-urban poultry farms in Dakar and Thiès, in Senegal. The domestic poultry industry in Senegal is rapidly growing, and is a key user of antibiotics [10,11]. Semi-intensive farms were selected because they comprise a very large portion of agricultural production in Senegal and many other middle-income countries, the group of countries which is most vulnerable to the effects of AMR [12,13]. In other countries, the shift from backyard farming to small- and medium-sized semi-intensive farms in recent decades has been associated with a range of novel and diverse farming practices [14]; in some cases meaning more indiscriminate antibiotic use [15,16], with medium-sized farms especially likely to misuse antibiotics [17]. Semi-intensive farms are also more economically vulnerable than larger-scale farms, and may have a precarious relationship to creditors and suppliers [16], making them a key target for this investigation. In Senegal, while many studies have been carried out on AMU in poultry farms, these studies tend to be descriptive and focus on mapping out knowledge, attitudes, and practices (KAP). This is the first study of this kind pointing to evidence on interventions to reduce AMU in Senegal.

We aimed to investigate factors which could induce farmers to reduce antibiotic use, guide more prudent use, or guard against productivity losses in the event of an antibiotic use reduction intervention. We identified three such factors to investigate, namely: (1) vaccination of chickens; (2) farmers’ attitudes to, and awareness of, AMR; and (3) on-farm biosecurity measures. We hypothesised that all three could lead to lower and better-informed AMU and/or could enhance productivity, reducing the need for antibiotics as growth promotion and disease management tools.

Using survey data collected with a modified AMUSE survey tool [18] from 222 farms in Dakar and Thiès, we investigated:Whether better biosecurity, vaccination, and awareness of AMR lead to lower or more selective use of antibiotics (e.g., limiting use to therapeutic use, or avoiding use of antibiotics intended for use in humans) in poultry farms.What effect these three factors, as well as antibiotic use (defined by expenditure on antibiotics), have on farm profitability and disease incidence.

Following our main results, we also investigate how these factors interact with each other, and explored additional specifications. 

## 2. Results

### 2.1. Descriptive Statistics

Of the 222 farms in our dataset, 124 had broilers only and 97 had layers only, with one farm having both. 

Table 1 (below) shows the distribution of categorical variables, and Figure 1 shows the distribution of continuous variables. Correlations (Pearson’s correlation coefficient) between key variables are displayed in Appendix F.

Histograms of the size of broiler and layer farms, and box-and-whisker plots showing the distribution of key variables used.

### 2.2. Main Results

Table 2, Table 3, Table 4, Table 5 and Table 6 (below) shows the results of our main regressions, where we look at the effect of our three main covariates (“biosecurity”, “AMR attitudes”, and “Vaccination”) on the quantity of AMU (“AMU quantity”) (Table 2); the likelihood of using antibiotics on healthy birds (“AMU in healthy birds”) (Table 3); animal morbidity (“Disease incidence”) (Table 4); and farm productivity (“broiler productivity” and “layer productivity”) (Table 5 and Table 6). 

None of our covariates of interest significantly affected the quantity of AMU, regardless of whether they were included together or separately (Table 2). In fact, farm size and production type were the only variables that significantly influenced this, with larger farms consistently using fewer antibiotics per bird (perhaps due to economies of scale) and broiler farms using less per cycle (although production cycles were much shorter). 

In univariate specifications (Appendix B), farmers with more ‘AMR-aware’ attitudes, and those with better biosecurity, appeared less likely to use antibiotics on healthy birds. However, there is little evidence to support the link with biosecurity in our main specifications (Table 3). Here, AMR-aware attitudes remained negatively associated with antibiotics use in healthy birds, but this relationship was not quite statistically significant (*p* = 0.113 and *p* = 0.120). Broiler farms were consistently more likely to use antibiotics in healthy birds, perhaps due to growth-promotion use.

Antibiotic use was consistently associated with a higher incidence of disease (Table 4), as was our index of vaccination (in the univariate specification only). We speculate that this reflects endogeneity in two ways: (1) that vaccination and antibiotics may be used in response to disease outbreaks; and (2) that farmers who are more aware of animal health are both more likely to report disease incidence and also more likely to vaccinate. 

A larger farm size, greater use of antibiotics, and better biosecurity were associated with more productive broilers (Table 5). In the univariate specifications (Appendix B), better vaccination was also associated with higher broiler productivity. Although antibiotics seemed to increase broiler productivity, so did vaccination and biosecurity. Therefore, a reduction in AMU with a simultaneous improvement in biosecurity (and vaccination) could improve antibiotic stewardship on broiler farms without harming productivity. This does not seem to be the case for layer farms, where none of our covariates significantly predicted productivity (Table 6).

### 2.3. Robustness

Following our main results, we regressed the quantity of AMU against each of the biosecurity measures individually, as opposed to the biosecurity index (“Biosecurity”) used elsewhere. Only four individual measures appeared to be significant, but they did not remain significant after adjusting for the false discovery rate or the family-wise error rate.

We also investigated the effect of having a relevant professional (veterinarian, paraveterinarian, or livestock helper) advise on antimicrobial use, on the quantity of AMU and the likelihood of using antimicrobials in healthy birds (Appendix G). However, this did not significantly affect either outcome.

After this, we used Heckman selection [19] to take account of farms which did not use antibiotics. Our covariates of interest had no significant effect, which is unsurprising given that only 13 farms (9.7%) out of 134 with data on antibiotic expenditure reported zero expenditure.

Finally, we examined three sub-hypotheses using interaction terms, all with disease incidence and productivity as our outcomes of interest (Appendix C). (1) We interacted AMU with biosecurity to see if better biosecurity reduced the need for antibiotics in improving farm outcomes. (2) We interacted vaccination and biosecurity to see if these two measures are substitutes in terms of disease management. (3) We interacted AMR attitudes with AMU to see if better awareness of AMR increased the effectiveness of antibiotics as a disease management tool (following our original assumption that AMU would have a negative effect on disease incidence). However, none of the interaction terms were statistically significant.

## 3. Discussion

### 3.1. Overview of Findings

The characteristics and production type of farms were shown to be just as important to antibiotic use practices and farm outcomes, as were our covariates of interest (biosecurity, vaccination, and AMR attitudes). Larger farms consistently used fewer antibiotics per bird, and had more productive broilers. Broiler farms also seemed more likely to use antibiotics on healthy birds. This could be explained by the fact that broiler production cycles are short with farmers desiring quick turn over, as farmers may wish to speed up production cycles using antibiotic growth promotors. Antibiotic use did seem to be associated with a greater productivity in broilers, but not in layers, suggesting a possible growth-promoting role.

Farmers with more ‘AMR-aware’ attitudes were less likely to use antibiotics on healthy birds in some specifications, which can be seen as indicative of more prudent AMU.

Vaccination was associated with more productive broilers in some specifications, and may be endogenous with disease incidence. Of our three covariates of interest, vaccination likely requires further investigation the most, due to the low variation in vaccination practices among the farms surveyed. This also means that vaccination may have effects that we were not able to capture in this study.

Biosecurity, as measured by an index of various farm practices, was associated with more productive broilers. In univariate specifications, it was also associated with a lower likelihood of using antibiotics in healthy birds. 

### 3.2. Comparison with Previous Work

Previous studies using the AMUSE tool have focused on characterising farm KAP. Our addition of questions concerning productivity, biosecurity, vaccination, and attitudes and knowledge of AMR greatly enhance the tool. This version of the survey (Appendix A) can also be used in other contexts, and a replication of our results in other contexts would yield very useful comparisons.

While the effectiveness of antibiotic growth promoters is controversial, there are reasons to believe that low (sub-inhibitory) doses can promote livestock productivity [9]. Our findings suggest that this may be the case, at least for semi-intensive broiler farms. This reaffirms the necessity of finding interventions which make antibiotic use reduction safer for farmers.

Weaker biosecurity infrastructure has also been associated with worse disease outcomes in other contexts [20]. We did not replicate this result, but we did find a link to broiler productivity.

Lastly, vaccination of poultry is potentially a very effective tool for productivity enhancement and disease management [21]. We found some suggestion of a productivity benefit for broilers, but did not replicate this finding consistently, likely reflecting the small sample size and very low variation in vaccination practices observed. 

### 3.3. Meaning of Results and Implications for Future Research

Overall, there is some evidence that our three factors of interest (biosecurity, vaccination, and AMR attitudes) could be used to reduce AMU in poultry production, either by modulating AMU behaviours or by mitigating the potential productivity lost due to antibiotic withdrawal. Specifically, biosecurity may lower the incidence of disease and reduce the need for therapeutic antibiotic use, and biosecurity and vaccination may offset any productivity loss associated with antibiotic use reduction. In addition, awareness raising and biosecurity improvements may reduce the use of antibiotics in healthy birds and improve prudence to antibiotic use.

The findings aim to inform key interventions of the next multisectoral AMR monitoring action plan for Senegal (2023–2027). The previous plan lasted 5 years and ended in 2022. The overall objective of the plan is to provide an effective response, through an integrated “One Health” approach, to the growing threat of antimicrobial resistance. Specific objectives of the plan which these results can inform include ensuring rational management and use of antimicrobials; informing and raising awareness on the issue of antimicrobial resistance; and the rational use of antimicrobials in animal health. 

### 3.4. Limitations

Using observational survey data such as these poses a few difficulties. For one, there was considerable endogeneity between antibiotic use, vaccination, and disease prevalence, which made causality difficult to disentangle. We recommend the use of larger datasets and annual follow-up to improve the statistical power of this type of study, as well as the use of instrumental variable techniques to mitigate this endogeneity. In particular, more data on the effectiveness of animal vaccination in semi-intensive poultry farms is necessary. Beyond this, a key step to follow should be to test these observational findings in the context of farm-level trials. Antibiotic use reduction (or replacement by non-antimicrobial feed additives) should be trialled alone, as well as in combination with interventions related to vaccination, biosecurity, and awareness raising. Outcomes measured should include the incidence of disease, farm productivity, the use of antibiotics, the level of resistance in livestock animals, and the extent to which farmers feel safe and willing to withdraw antibiotics.

The relative homogeneity of farms in terms of practices (for example, near-universal antibiotic use and consistent vaccine coverage) not only contrasted stylised facts about the diversity and inconsistency of semi-intensive poultry farming practices [14], but also made statistical inference more difficult. This reaffirms the potential use of farm-level trials, in which these variables are intentionally altered, in future research. We were also not able to obtain enough detail about the different types of vaccines used to investigate this as a factor at this sample size. Since the types of vaccines used will vary among farms, it would be important to understand the differential effect of each vaccine on animal health and antibiotic usage when informing policy.

There was also a very low R^2^ value across all regression specifications, likely reflecting the omission of key variables. A more detailed understanding of the relevant production system, for example using system dynamic models informed and parameterised in consultation with stakeholders [22], could help to collect more relevant data and to build more relevant models. Along with colleagues, we have recently submitted a paper which uses stakeholder elicitation to build a system dynamic model of this production system, which investigates the relative importance of potential interventions targeting AMU and profitability.

While we investigated the effect of awareness and attitudes from a statistical perspective, this is not a substitute for an in-depth investigation of these attitudes using mixed-methods research. Other upcoming research using this modified survey tool aims to answer this question in greater detail.

A further limitation is that we were not able to collect data on the actual quantity of antibiotics used, e.g., in defined daily doses, as these data were not collected by farmers, and instead we had to use expenditure on antibiotics as a proxy. This may have introduced bias due to the different prices of various antibiotic types, meaning that these results are harder to compare directly with those from other contexts (or to other metrics such as the global average annual consumption of antibiotics).

Finally, it must be noted that these findings alone may not be sufficient to facilitate changes to farming practices. The adoption of better biosecurity and vaccination practices are not a matter of individual ‘smart choices’, but are structural and nationwide issues that are heavily dependent on infrastructure and state support, being more effective when rolled out nationally [20,21]; attitudes towards AMR can be thought of in the same way. Farmers moving towards more intensified production systems are exposed to novel challenges and require appropriate state support [14]. Semi-intensive farmers often exist in a state of financial precarity and may require systems of financial support, such as insurance, in order to feel safe altering their practices. Small- and medium-scale farmers have complex upstream and downstream relationships with actors, such as suppliers and creditors [16]. Farmers must be seen as a part of this network, rather than as individual actors, and stakeholders from across this system must be meaningfully consulted in the formulation of future research and interventions.

## 4. Materials and Methods

### 4.1. Study Aims, Data Collection Methods, and Setting

The data used in this study came from a modified version of the AMUSE survey tool, which is used to explore farm characteristics and AMR KAP in livestock farms [18]. The original AMUSE tool has been used for descriptive purposes in Senegal as well as in other country settings, and our adapted version was expanded to include more measures of farm productivity, antibiotic use quantity, antibiotic use prudence, vaccination of livestock, AMR knowledge and attitudes, and on-farm biosecurity practices. ‘Prudence’ in this study refers to the use/non-use of antibiotics in healthy birds. Also of relevance is the effect that these variables have on farm productivity and disease incidence in chickens. Data were cleaned and answers from sets of questions were compiled into metrics and indices for easier analysis (Table 7).

The locations of the farms surveyed are detailed in Figure 2 (below). Data collection took place from February to September 2022. A snowball sampling method was used to select farms. This method was chosen because a national database of poultry producers has not yet been compiled, making other sampling methods prohibitively difficult. A representative from each farm was interviewed for an average of one hour per farm. Four people in total were responsible for data collection, divided into two pairs, each composed of a veterinary doctoral student and a member of the Veterinary Service Division (DSV) of Dakar or Thiès, sometimes with the addition of a livestock technician to act as a guide and interlocutor. Data were collected electronically on smartphones using the Open Data Kit (ODK) platform.

The map above covers the six departments in which surveyed farms were located, with location within Senegal indicated in the top-right corner. Each red point represents the location of one of the 222 farms which were surveyed for the purposes of this study.

The full set of survey questions used can be found in Appendix A. Ethical approval can be found in Appendix D and a translated copy of the informed consent form used for the study can be found in Appendix E. Being an observational study, this paper conforms to the Strengthening the Reporting of Observational Studies in Epidemiology (STROBE) checklist [23].

### 4.2. Variables Used

We first present descriptive statistics, and then use regression analysis to look at the association between our three variables of interest on the following outcomes (both univariate and controlling for farm characteristics): quantity of AMU (measured by expenditure per bird), the use of antibiotics on healthy birds, farm productivity (meat and egg production), and the incidence of disease. Table 3 (below) details the variable names used throughout this paper, and outlines how variables were derived where relevant. 

Farm characteristics that were controlled for included the ratio of broilers to layers, farm size, and the presence or absence of livestock species other than chickens. This is because broiler and layer farms have different production stages and may use antibiotics in different ways; farm size may influence access to resources and economies of scale, and the presence of other species raises additional concerns of cross-contamination and may affect the efficacy of vaccination [21].

### 4.3. Main Statistical Methods

All statistical analyses were carried out using R version 4.1.2 [24] via RStudio build 554 [25]. Key packages used include Stargazer [26], Tidyverse [27], ggplot2 [28], Corrplot [29], and dplyr [30]. Model specifications were not chosen based on explanatory power (e.g., AIC or BIC) but were pre-specified in the pre-analysis plan based on theory. This is because we wanted to test specific hypotheses about our chosen variables rather than simply finding the model with the greatest explanatory power. Alternative specifications were explored during robustness testing.

First, we regressed the quantity of antibiotics used (“AMU quantity”) against each of the three main covariates using ordinary least squares (OLS) (models (1)–(3)), and then against all three main covariates together (model (4)). We adjusted for key farm characteristics of farm size, presence of other species, and the ratio of broilers to layers. We then did this for other outcomes, namely disease incidence and productivity (“broiler productivity” and “layer productivity”).
(1)$$Y_{t}=\beta_{0}+\beta_{1}*vaccination+\beta_{2}*farm\;size+\beta_{3}*other\;species+\beta_{4}*portion\;broilers+\varepsilon$$
(2)$$Y_{t}=\beta_{0}+\beta_{1}*biosecurity+\beta_{2}*farm\;size+\beta_{3}*other\;species+\beta_{4}*portion\;broilers+\varepsilon$$
(3)$$Y_{t}=\beta_{0}+\beta_{1}*AMR\;attitudes+\beta_{2}*farm\;size+\beta_{3}*other\;species+\beta_{4}*portion\;broilers+\varepsilon$$
(4)$$Y_{t}=\beta_{0}+\beta_{1}*vaccination+\beta_{2}*biosecurity+\beta_{3}*AMR\;attitudes+\beta_{4}*farm\;size+\beta_{5}*other\;species+\beta_{6}*portion\;broilers+\varepsilon$$
where i ϵ {AMU quantity, disease incidence, broiler productivity, layer productivity}.

Aside from wanting to investigate the determinants of AMU, we also looked at disease incidence and productivity to see if the three measures of interest (vaccination, biosecurity, and awareness raising) incur any trade-offs in terms of profitability. Ultimately, if we recommend these measures as means of encouraging farmers to reduce or modulate AMU, then we should be confident that this will not endanger their economic security or broader food security at the population level.

Following this, we regressed use of antibiotics on healthy birds against each of the three main categories of covariates using a logistic regression (logit). These logistic regressions were performed in order to see if any of the three measures being investigated improved prudent use of antibiotics.
(5)$$p={\left(1+e^{-xb}\right)}^{-1}$$
where p is the probability of using antibiotics in healthy birds; x is a vector of the covariates used in models (1) through (4); and b is a vector of parameters (odds ratio).

### 4.4. Robustness and Further Specifications

We first tested the association between AMU and a large number of individual biosecurity measures, as opposed to the biosecurity index (“biosecurity”) used elsewhere. We accounted for multiple hypothesis testing using family-wise error rate (using Bonferroni correction [31]) and the false discovery rate (using the Benjamini-Hochberg step-up procedure [32]).

After this, we looked at the effect of our three main covariates on AMU using Heckman selection [19], with a selection function using variables that were seen to affect AMU in other specifications. This was done to take better account of farms which did not use antibiotics.

Finally, we looked at a number of interactions between key covariates to test more specific hypotheses. All of these interactions looked at productivity and animal mortality as outcomes. The hypotheses are described below.

Interacting AMU with biosecurity to see if better biosecurity reduced the need for antibiotics in improving farm outcomes, i.e.,
(6)$$broiler\;productivity=\beta_{0}+\beta_{1}*biosecurity+\beta_{2}*AMU\;quantity+\beta_{3}*biosecurity*AMU\;quantity+\beta_{4}*farm\;size+\beta_{5}*other\;species+\varepsilon$$
(7)$$layer\;productivity=\beta_{0}+\beta_{1}*biosecurity+\beta_{2}*AMU\;quantity+\beta_{3}*biosecurity*AMU\;quantity+\beta_{4}*farm\;size+\beta_{5}*other\;species+\varepsilon$$
(8)$$disease\;incidence=\beta_{0}+\beta_{1}*biosecurity+\beta_{2}*AMU\;quantity+\beta_{3}*biosecurity*AMU\;quantity+\beta_{4}*farm\;size+\beta_{5}*other\;species+\beta_{5}*portion\;broilers+\varepsilon$$

Interacting vaccination and biosecurity to see if these two measures are substitutes in terms of disease management, i.e.,
(9)$$broiler\;productivity=\beta_{0}+\beta_{1}*biosecurity+\beta_{2}*vaccination+\beta_{3}*biosecurity*vaccination+\beta_{4}*farm\;size+\beta_{5}*other\;species+\varepsilon$$
(10)$$layer\;productivity=\beta_{0}+\beta_{1}*biosecurity+\beta_{2}*vaccination+\beta_{3}*biosecurity*vaccination+\beta_{4}*farm\;size+\beta_{5}*other\;species+\varepsilon$$
(11)$$disease\;incidence=\beta_{0}+\beta_{1}*biosecurity+\beta_{2}*vaccination+\beta_{3}*biosecurity*vaccination+\beta_{4}*farm\;size+\beta_{5}*other\;species+\beta_{5}*portion\;broilers+\varepsilon$$

Interacting AMR attitudes with AMU to see if better awareness of AMR increased the effectiveness of antibiotics as a disease management tool (following the hypothesis that AMU will be negatively associated with disease incidence), i.e.,
(12)$$broiler\;productivity=\beta_{0}+\beta_{1}*AMR\;attitudes+\beta_{2}*AMU\;quantity+\beta_{3}*AMR\;attitudes*AMU\;quantity+\beta_{4}*farm\;size+\beta_{5}*other\;species+\varepsilon$$
(13)$$layer\;productivity=\beta_{0}+\beta_{1}*AMR\;attitudes+\beta_{2}*AMU\;quantity+\beta_{3}*AMR\;attitudes*AMU\;quantity+\beta_{4}*farm\;size+\beta_{5}*other\;species+\varepsilon$$
(14)$$disease\;incidence=\beta_{0}+\beta_{1}*AMR\;attitudes+\beta_{2}*AMU\;quantity+\beta_{3}*AMR\;attitudes*AMU\;quantity+\beta_{4}*farm\;size+\beta_{5}*other\;species+\beta_{5}*portion\;broilers+\varepsilon$$

## 5. Conclusions

We did not find consistent evidence that biosecurity, vaccination, and attitudes towards AMR reduce the overall quantity of AMU or the use of antimicrobials in healthy birds, although better biosecurity and “AMR-aware” attitudes were associated with less use in healthy birds in some specifications. However, we did find evidence that biosecurity, and potentially vaccination, could mitigate the risks of antibiotic withdrawal in broiler farms by improving productivity.

These findings should be explored further using annual follow-up, larger sample sizes, and farm-level trials which combine antibiotic withdrawal and replacement with interventions in these three areas. Finally, these findings alone may not be sufficient to catalyse change in agricultural stewardship of antimicrobials. AMR in agriculture must always be seen as a structural rather than an individual issue, with stakeholders from across the One Health spectrum meaningfully consulted as part of research and policymaking.

## Figures and Tables

**Figure 1 antibiotics-12-00460-f001:**
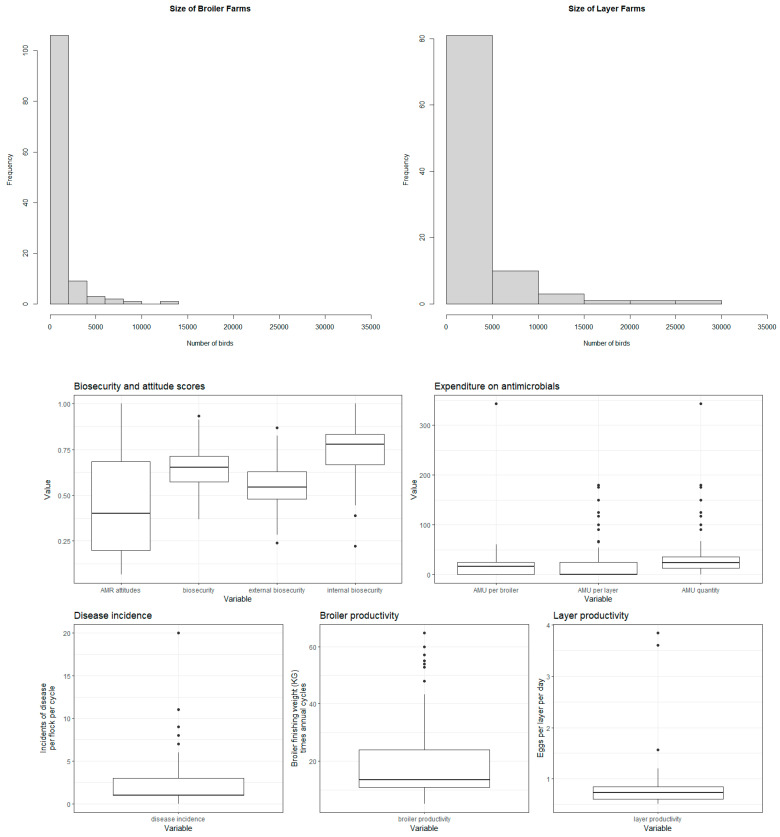
Distribution of continuous variables.

**Figure 2 antibiotics-12-00460-f002:**
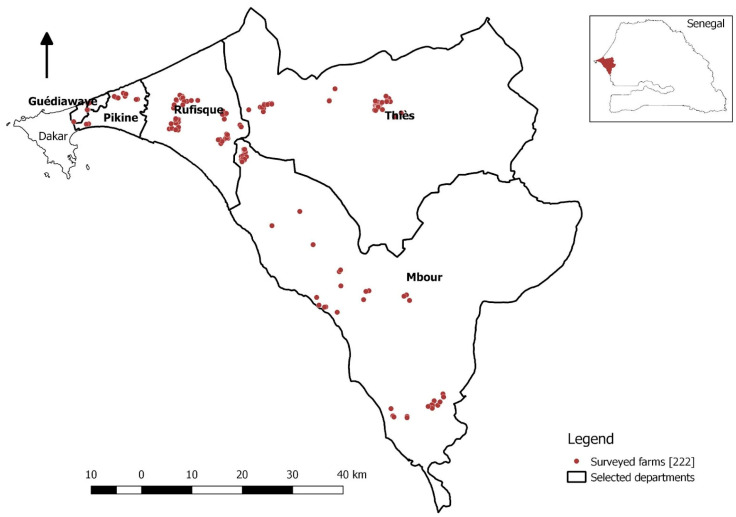
Map of the farms surveyed and the selected area.

**Table 1 antibiotics-12-00460-t001:** Summary statistics of categorical variables.

Variable	Description
“Vaccination”	no protocol (15/222) protocol in place but not always adhered to (6/222) protocol in place and always adhered to (201/222)
“Other species on farm”	no species present other than chickens (193/222) other species present (29/222)
“AMU in healthy birds”	did not use antibiotics in healthy birds (216/222) used antibiotics in healthy birds (6/222)
“Portion broilers”	Broilers only: 124/222 Layers only: 97/222 53% broilers and 47% layers: 1/222

Summary statistics of categorical variables used.

**Table 2 antibiotics-12-00460-t002:** Determinants of AMU quantity.

*Dependent Variable*				
	“AMU quantity”			
	(1)	(2)	(3)	(4)
vaccination	4.130			5.545
	(5.925)			(6.254)
biosecurity		−10.110		−10.815
		(33.572)		(36.783)
“AMR attitudes”			−8.173	−9.224
			(12.762)	(13.718)
“farm size”	−0.003 **	−0.003 **	−0.003 **	−0.003 **
	(0.001)	(0.001)	(0.001)	(0.001)
“other species on farm”	−3.767	−3.100	−3.585	−3.981
	(9.838)	(9.860)	(9.829)	(9.934)
“portion broilers”	−17.822 **	−18.575 **	−18.197 **	−18.156 **
	(7.439)	(7.552)	(7.424)	(7.593)
Constant	41.024 ***	55.206 **	52.623 ***	49.912 **
	(12.857)	(22.799)	(9.176)	(23.493)
Observations	134	134	134	134
R^2^	0.071	0.068	0.070	0.076
Adjusted R^2^	0.042	0.039	0.042	0.033
Residual Std. Error	39.553 (df = 129)	39.614 (df = 129)	39.565 (df = 129)	39.751 (df = 127)
F Statistic	2.464 ** (df = 4; 129)	2.358 * (df = 4; 129)	2.444 ** (df = 4; 129)	1.746 (df = 6; 127)

Note: * *p* < 0.1; ** *p* < 0.05; *** *p* < 0.01; (1) Effect of vaccination on AMU; (2) effect of biosecurity on AMU; (3) effect of attitudes on AMU; and (4) effect of all three on AMU (standard errors in parentheses).

**Table 3 antibiotics-12-00460-t003:** Determinants of antibiotic use in healthy birds.

*Dependent Variable*				
	“AMU in healthy birds”			
	(1)	(2)	(3)	(4)
vaccination	0.077			0.137
	(0.290)			(0.300)
biosecurity		−0.326		0.163
		(1.520)		(1.602)
“AMR attitudes”			−0.846	−0.896
			(0.543)	(0.565)
“farm size”	−0.00005	−0.00005	−0.00004	−0.00005
	(0.0001)	(0.0001)	(0.0001)	(0.0001)
“other species on farm”	0.009	0.019	0.017	0.005
	(0.443)	(0.443)	(0.446)	(0.447)
“portion broilers”	1.606 ***	1.582 ***	1.600 ***	1.621 ***
	(0.314)	(0.322)	(0.315)	(0.327)
Constant	−0.655	−0.298	−0.117	−0.456
	(0.606)	(1.039)	(0.372)	(1.089)
Observations	220	220	220	220
Log Likelihood	−132.635	−132.647	−131.451	−131.329
Akaike Inf. Crit.	275.271	275.294	272.902	276.659

Note: *** *p* < 0.01; (1) Effect of vaccination on antibiotic use in healthy birds; (2) effect of biosecurity on antibiotic use in healthy birds; (3) effect of attitudes on antibiotic use in healthy birds; and (4) effect of all three on antibiotic use in healthy birds (standard errors in parentheses).

**Table 4 antibiotics-12-00460-t004:** Determinants of disease incidence.

*Dependent Variable*				
	“disease incidence”			
	(1)	(2)	(3)	(4)
vaccination	0.409			0.515
	(0.327)			(0.344)
biosecurity		−0.251		−0.199
		(1.858)		(2.016)
“AMR attitudes”			−0.794	−1.000
			(0.705)	(0.753)
“AMU quantity”	0.011 **	0.012 **	0.012 **	0.011 **
	(0.005)	(0.005)	(0.005)	(0.005)
“farm size”	−0.00003	−0.00003	−0.00002	−0.00002
	(0.0001)	(0.0001)	(0.0001)	(0.0001)
“other species on farm”	−0.167	−0.116	−0.148	−0.208
	(0.542)	(0.546)	(0.542)	(0.545)
“portion broilers”	−0.564	−0.603	−0.600	−0.575
	(0.419)	(0.427)	(0.419)	(0.425)
Constant	1.481 **	2.381 *	2.618 ***	1.923
	(0.736)	(1.290)	(0.567)	(1.310)
Observations	134	134	134	134
R^2^	0.084	0.073	0.082	0.098
Adjusted R^2^	0.048	0.036	0.046	0.048
Residual Std. Error	2.179 (df = 128)	2.192 (df = 128)	2.181 (df = 128)	2.178 (df = 126)
F Statistic	2.337 ** (df = 5; 128)	2.004 * (df = 5; 128)	2.274 * (df = 5; 128)	1.959 * (df = 7; 126)

Note: * *p* < 0.1; ** *p* < 0.05; *** *p* < 0.01; (1) Effect of vaccination on disease incidence; (2) effect of biosecurity on disease incidence; (3) effect of attitudes on disease incidence; and (4) effect of all three on disease incidence (standard errors in parentheses).

**Table 5 antibiotics-12-00460-t005:** Determinants of productivity (broilers).

*Dependent Variable*				
	“broiler productivity”			
	(1)	(2)	(3)	(4)
vaccination	2.477			0.847
	(2.450)			(2.573)
biosecurity		28.393 **		24.638
		(13.198)		(15.179)
“AMR attitudes”			7.423	2.330
			(5.815)	(6.458)
“AMU quantity”	0.121 ***	0.121 ***	0.126 ***	0.120 ***
	(0.041)	(0.040)	(0.040)	(0.040)
“farm size”	0.004 ***	0.004 ***	0.005 ***	0.004 ***
	(0.001)	(0.001)	(0.001)	(0.001)
“other species on farm”	2.364	2.174	3.491	2.388
	(4.151)	(4.056)	(4.173)	(4.196)
Constant	8.087 *	−4.919	8.048 **	−5.264
	(4.628)	(8.247)	(3.995)	(8.484)
Observations	84	84	84	84
R^2^	0.214	0.248	0.220	0.251
Adjusted R^2^	0.174	0.210	0.180	0.192
Residual Std. Error	13.641 (df = 79)	13.344 (df = 79)	13.590 (df = 79)	13.492 (df = 77)
F Statistic	5.378 *** (df = 4; 79)	6.511 *** (df = 4; 79)	5.569 *** (df = 4; 79)	4.292 *** (df = 6; 77)

Note: * *p* < 0.1; ** *p* < 0.05; *** *p* < 0.01; (1) Effect of vaccination on broiler productivity; (2) effect of biosecurity on broiler productivity; (3) effect of attitudes on broiler productivity; and (4) effect of all three on broiler productivity (standard errors in parentheses).

**Table 6 antibiotics-12-00460-t006:** Determinants of productivity (layers).

*Dependent Variable*				
	“layer productivity”			
	(1)	(2)	(3)	(4)
vaccination	−0.072			−0.082
	(0.062)			(0.064)
biosecurity		−0.357		−0.335
		(0.417)		(0.460)
“AMR attitudes”			−0.106	−0.076
			(0.084)	(0.092)
“AMU quantity”	−0.001	−0.001	−0.001	−0.001
	(0.001)	(0.001)	(0.001)	(0.001)
“farm size”	−0.00000	0.00000	−0.00000	0.00000
	(0.00000)	(0.00001)	(0.00000)	(0.00001)
“other species on farm”	0.115	0.119	0.096	0.112
	(0.069)	(0.071)	(0.070)	(0.072)
Constant	0.876 ***	0.956 ***	0.797 ***	1.149 ***
	(0.127)	(0.263)	(0.062)	(0.326)
Observations	26	26	26	26
R^2^	0.189	0.166	0.198	0.268
Adjusted R^2^	0.035	0.007	0.045	0.036
Residual Std. Error	0.112 (df = 21)	0.113 (df = 21)	0.111 (df = 21)	0.112 (df = 19)
F Statistic	1.226 (df = 4; 21)	1.043 (df = 4; 21)	1.295 (df = 4; 21)	1.157 (df = 6; 19)

Note: *** *p* < 0.01; (1) Effect of vaccination on layer productivity; (2) effect of biosecurity on layer productivity; (3) effect of attitudes on layer productivity; and (4) effect of all three on layer productivity (standard errors in parentheses).

**Table 7 antibiotics-12-00460-t007:** Variable glossary.

Varname	Meaning	Units
“Vaccination”	A score for having (and adhering to) a vaccination protocol for birds	0 = no protocol 1 = protocol in place but not always adhered to 2 = protocol in place and always adhered to
“Internal biosecurity”	A score for internal biosecurity measures on the farm, based on categorical responses to several questions about biosecurity procedures. As with other scores in this dataset, each question about internal biosecurity gave a number of points (1 for the ‘best’ answer and 0 for the ‘worst’ answer, with fractions for answers in between). The internal biosecurity score is then calculated as the mean of the scores attained on all questions about internal biosecurity	Continuous, ranging between 0 (met none of the standards) and 1 (met every standard)
“External biosecurity”	A score for external biosecurity measures on the farm, based on categorical responses to several questions about biosecurity procedures	Continuous, ranging between 0 (met none of the standards) and 1 (met every standard)
“Biosecurity”	The mean of the internal and external biosecurity scores	Continuous, ranging between 0 (met none of the standards) and 1 (met every standard)
“AMR attitudes”	A score for attitudes about antimicrobial resistance and stewardship, based on categorical responses to several questions	Continuous, ranging between 0 (met none of the standards) and 1 (met every standard)
“Farm size”	The number of chickens on the farm	Chickens
“Other species on farm”	The presence of animal species other than chickens on the farm	Binary 0 = no other species present 1 = other species present
“AMU quantity”	The quantity of antibiotics used in chicken production. “AMU per broiler” and “AMU per layer” disaggregate this figure by production type	FCFA (Franc de la Communauté Financière Africaine, or West African Franc) spent on antibiotics per bird per production cycle
“AMU in healthy birds”	The use of antibiotics in healthy birds (for whatever reason)	Binary 0 = did not use antibiotics in healthy birds 1 = used antibiotics in healthy birds
“Disease incidence”	Amount of disease occurring in the flock	Number of individual disease incidents recorded in the flock during a production cycle
“Broiler productivity”	Productivity of broilers	Average finishing weight of a broiler, multiplied by the number of production cycles per year
“Layer productivity”	Productivity of layers	Average number of eggs laid per layer per day
“Portion broilers”	Portion of chickens on the farm which are broilers	Portion

## Data Availability

The data used in this study, as well as the code used for data cleaning and analysis, are available in anonymised form on GitHub at https://github.com/Trescovia/AMUSE-SEFASI-Sharing (accessed on 23 February 2023). The repository is password-protected, and will be made available upon request to those who email the corresponding author and provide a reason for their desire to access it.

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
