# Peer review of "Drivers of Antibiotic Use in Semi-Intensive Poultry Farms: Evidence from a Survey in Senegal"

_antibiotics, 2023, doi:10.3390/antibiotics12030460_

Round 1
Reviewer 1 Report
The paper deals with an important and interesting topic that is directly related to ΟΝΕ HEALTΗ. The design of the experiment is complete and the questionnaire records all possible parameters involved and influencing the use of antibiotics in poultry farms.
I believe that the processing of the results has the following weaknesses
1. The consumption of antibiotics is not fully analyzed (global average annual consumption of antibiotics).
2. The possible difference in the use of antibiotics between informed and non-informed poultry farmers is not mentioned.
3. Antibiotics, which are widely used mainly without a prescription for preventive reasons or as growth factors, are not mentioned.
4. Last but not least, the vaccines used and to what extent they affect the reduction of antibiotic use
I think that when these data are added in both results and discussion, the paper will be accepted for publication.
Author Response
Thank you for your comments! We have endeavoured to incorporate them all into the manuscript, and have detailed our responses to each comment below (changes made are highlighted)
The paper deals with an important and interesting topic that is directly related to ΟΝΕ HEALTΗ. The design of the experiment is complete and the questionnaire records all possible parameters involved and influencing the use of antibiotics in poultry farms.
Thank you very much!
I believe that the processing of the results has the following weaknesses
1) The consumption of antibiotics is not fully analyzed (global average annual consumption of antibiotics.
We were not able to analyse this directly as farmers did not record the DDD used. we have added this as a limitation (lines 335-340)
2) The possible difference in the use of antibiotics between informed and non-informed poultry farmers is not mentioned.
We have added specifications testing the effect of having AMU advised by a professional (vet, paravet, or livestock helper) on the quantity of AMU and the likelihood of using antimicrobials in healthy birds (lines 211-214 and Appendix G)
3) Antibiotics, which are widely used mainly without a prescription for preventive reasons or as growth factors, are not mentioned.
We have expanded on this in lines 59-62
4) Last but not least, the vaccines used and to what extent they affect the reduction of antibiotic use
Unfortunately, farmers were not able to give us enough detail about vaccines used to analyse the effects of different vaccine types with this sample size, which we have added as a limitation in lines 316-319. We investigate the effect of vaccination on antibiotic use in table C-1
I think that when these data are added in both results and discussion, the paper will be accepted for publication.
Reviewer 2 Report
In the presented manuscript, Emesand co-workers analyzed “Drivers of antibiotic use in semi-intensive poultry farms: evidence from a survey in Senegal”.
The manuscript is suitable for publication in Antibiotics, the language should be revised and additional analyses should be performed to present a comprehensive analysis as proposed by the authors.
I provide some major and minor comments below.
Major;
1. Lines 286-290. Authors should limit the scope of the research.
2. Lines 313-317. The authors mention it but it is not applied. It would contribute a lot to the impact of the article if they included other variables that would allow them to obtain more robust results.
3. The conclusions are not in accordance with those stated at the end of the introduction:
Using survey data collected with a modified AMUSE survey tool(18) from 222 102 farms in Dakar and Thiès, we investigated: 103 a) If better biosecurity, vaccination and awareness of AMR lead to lower or more 104 selective use of antibiotics (e.g. limiting use to therapeutic use, or avoiding use 105 of antibiotics intended for use in humans) in poultry farms 106 b) What effect these three factors, as well as antibiotic use (defined by expenditure 107 on antibiotics), have on farm profitability and disease incidence
Report in points the conclusions based on what has been proposed.
Minor,
The writing and grammar of the article should be revised.
1. Figure A legend is not reported, the font size should be increased and the A1 size of broiler Farms graph should reduce the X-axis to 15,000.
2. It is impossible to read the information in the appendix, increase the resolution of: Appendix B – univariate specifications and Appendix C – specifications with interactions between our main covariates.
Author Response
Thank you very much for your comments! We have endeavoured to incorporate them all into the manuscript. We have detailed the changes made below (highlighted) with reference to each of your comments.
Comments:
In the presented manuscript, Emes and co-workers analyzed “Drivers of antibiotic use in semi-intensive poultry farms: evidence from a survey in Senegal”.
The manuscript is suitable for publication in Antibiotics, the language should be revised and additional analyses should be performed to present a comprehensive analysis as proposed by the authors.
I provide some major and minor comments below.
Major;
1) Lines 286-290. Authors should limit the scope of the research.
We have narrowed down the list of objectives to only the most relevant ones
2) Lines 313-317. The authors mention it but it is not applied. It would contribute a lot to the impact of the article if they included other variables that would allow them to obtain more robust results.
Lines 313-317 include:
"There was also a very low R2 value across all regression specifications, likely reflecting the omission of key variables. A more detailed understanding of the relevant production system, for example using system dynamic models informed and parameterised in consultation with stakeholders(22), could help to collect more relevant data and to build more relevant models"
In acknowledgment of this, we have recently submitted a paper which creates a system dynamic model of the poultry production system in this context, which investigates the relative importance of potential interventions. We have expanded our mention of this in lines 325-328
In terms of including more relevant variables, we have added some specifications which investigate the effect of professional AMU advice on the quantity and prudence of AMU (lines 211-214 and Appendix G). Data permitting the addition of further new variables were not collected.
3)The conclusions are not in accordance with those stated at the end of the introduction:
Using survey data collected with a modified AMUSE survey tool(18) from 222 102 farms in Dakar and Thiès, we investigated: 103
a) If better biosecurity, vaccination and awareness of AMR lead to lower or more 104 selective use of antibiotics (e.g. limiting use to therapeutic use, or avoiding use 105 of antibiotics intended for use in humans) in poultry farms 106
b) What effect these three factors, as well as antibiotic use (defined by expenditure 107 on antibiotics), have on farm profitability and disease incidence
Report in points the conclusions based on what has been proposed.
We have rephrased the conclusion to more closely follow the points made at the end of the introduction
Minor,
The writing and grammar of the article should be revised.
We have further proofread and edited the manuscript
1) Figure A legend is not reported, the font size should be increased and the A1 size of broiler Farms graph should reduce the X-axis to 15,000.
We have included and formatted a caption for this figure. We have kept the x-axes of both sides of this graph within the same domain to make them more easily comparable for the reader
2) It is impossible to read the information in the appendix, increase the resolution of: Appendix B – univariate specifications and Appendix C – specifications with interactions between our main covariates.
We have split these tables into several sections and made them easier to read
Reviewer 3 Report
I found this article interesting for the readers and followed the journal Antibiotics’ scope. I don’t have any major comments as this article has enough data and is well written with proper discussion.
I would recommend the article be published in Antibiotics after minor corrections.
The author needs to address the following comments/corrections.
1. The author should correct the format of references wherever needed (e.g Year Bold, Volume Italic etc).
2. Introduction needs to be shortened.
3. All the tables and figures should include footnotes.
4. Resolution of Figure B should be improved and needs to subdivide.
5. All Appendix should be in supporting Information.
Author Response
Thank you very much for your comments. We have endeavoured to incorporate all of them into our manuscript. The changes are detailed below (in italics)
I found this article interesting for the readers and followed the journal Antibiotics’ scope. I don’t have any major comments as this article has enough data and is well written with proper discussion.
Thank you very much!
I would recommend the article be published in Antibiotics after minor corrections.
The author needs to address the following comments/corrections.
1) The author should correct the format of references wherever needed (e.g Year Bold, Volume Italic etc).
We have gone through and added bold and italics to the references as appropriate
2) Introduction needs to be shortened.
We have shortened the introduction, focusing only on details directly relevant to the paper as well as the additions suggested by other reviewers
3) All the tables and figures should include footnotes.
We have added footnotes to tables and figures where possible
4) Resolution of Figure B should be improved and needs to subdivide.
We have increased the size and resolution of this figure
5) All Appendix should be in supporting Information.
We are uncertain what is meant by this. All supporting information was indeed included in the appendices
Round 2
Reviewer 1 Report
Dear authors
after your changes/additions/explanations your work is ready to be published.
Reviewer 3 Report
I would like to thank the authors for making necessary corrections and modifications to the manuscript as suggested by the reviewers which remarkably enhanced the overall quality of the manuscript with a clear presentation of results and proper discussion.
I would recommend the article be published in Antibiotics, after a minor revision. There are technical errors, I hope the editor will take care of them.
1. Check the font of “Table C-2 - determinants of antibiotic use in healthy birds” and wherever applicable.
2. Check the brightness of equations used in this manuscript.